# Potentials and Limitations of Subsidies in Sustainability Governance: The Example of Agriculture

Katharine Heyl [1,2,*], Felix Ekardt [1,3], Lennard Sund [4] and Paula Roos [1]

1    Research Unit Sustainability and Climate Policy, 04229 Leipzig, Germany
2    Faculty of Agricultural and Environmental Sciences, University of Rostock, 18051 Rostock, Germany
3    Faculty of Law and Interdisciplinary Faculty, University of Rostock, 18051 Rostock, Germany
4    Faculty of Management, Economics and Social Sciences, University of Cologne, 50923 Cologne, Germany
*    Correspondence: katharine.heyl@uni-rostock.de

**Abstract:** The goals of the Paris Agreement and the Convention on Biological Diversity call for a global transition to sustainability. To achieve these goals, subsidies can be implemented. Subsidies are pervasive especially (but not only) in the agricultural sector. The agricultural sector plays an important role in the transition to sustainability as agriculture can both benefit and harm climate and biodiversity. Some agricultural subsidies seem environmentally beneficial, but the majority appear environmentally destructive. Against this background, this article applies a qualitative governance analysis—including aspects of legal analysis—to provide a comprehensive review of agricultural subsidies in the EU and to discuss the role of subsidies in transitioning towards sustainability. Results show that agricultural subsidies need to be substantially downscaled and implemented as complementary instruments only because other policy instruments such as quantity control instruments are more effective in addressing the drivers of non-sustainability, i.e., fossil fuels and livestock farming. However, subsidies remain a useful complementary instrument to remunerate the provision of public goods (e.g., in nature conservation) as long as they are constructed in a way that they do not suffer from typical governance problems. In addition, data and transparency need to be improved, subsidies for research and development increased, and environmental objectives streamlined through EU law to ensure all agricultural subsidies are in line with global environmental goals.

**Keywords:** agricultural subsidies; subsidies; Common Agricultural Policy; Paris Agreement; Convention on Biological Diversity; State aid; Green Deal; Farm to Fork Strategy; public goods

## 1. Introduction

The list of political commitments to reform and phase out subsidies—including in agriculture—appears endless. Early on, the Brundtland Report highlighted the issues of (then persistent coupled) agricultural subsidies [1]. More recent examples include calls to phase out inefficient fossil fuel subsidies at the COP26 in Glasgow and by WTO members in 2021 [2] (para 20) [3]. However, reforming and effectively using subsidies appears difficult. For example, in 2011, the European Commission published its Roadmap to a Resource Efficient Europe. It aimed to phase out environmentally harmful subsidies by 2020 [4]. This milestone has never been achieved. Instead, fossil fuel subsidies have even increased since 2015 in eleven Member States [5].

Especially in agriculture, subsidies are pervasive. The OECD finds that countries around the globe subsidise the agricultural sector with USD 720 billion annually (2018–2020), which corresponds to approximately half of the EU budget for seven years (Council Regulation 2020/2093). Yet, in general, these subsidies largely fail to ensure food security, farm livelihoods and sustainable practices [6] (see also [7]). Agricultural subsidies affect environmental outcomes by changing how much is produced, which products are produced, where these products are produced, and how they are produced [8].



Despite the criticism, subsidies are not per se a policy instrument to be put in the corner. The aim of this article is to provide a comprehensive review of agricultural subsidies and to discuss the role of this policy instrument in transitioning the sector towards sustainability. To this end, a qualitative governance analysis is applied. We review the literature on subsidies in general and agricultural subsidies in particular, and combine it with an assessment of the legal framework in international environmental law and EU law. Although much literature has focussed on specific aspects of agricultural subsidies such as their impact on farm income, no study adopts a comprehensive perspective on potentials and limitations of subsidies in transitioning the agricultural sector towards sustainability (see also [9]). This article aims to fill this gap and argues that subsidies need to be substantially downscaled and implemented as a complementary instrument only because other policy instruments are more effective in addressing important drivers for unsustainability such as fossil fuels and livestock farming. Subsidies, however, remain a useful complementary instrument to remunerate the provision of certain public goods (e.g., in nature conservation) as long as they are designed in a way that they do not suffer from some typical governance problems. Besides these governance findings, legal obligations also point in this direction.

The article is structured as follows. Section 2 describes the qualitative governance analysis. Section 3 provides the results in two parts. The first part analyses the characteristics of subsidies, including informal subsidies, and provides a legal definition of subsidies. This part also analyses the general potentials and limitations of subsidies and provides an overview of subsides in the agricultural sector. The second part of Section 3 introduces the normative framework of subsidies including international environmental treaties and EU provisions. Section 4 is a discussion focusing on the reform and removal of agricultural subsidies, (lacking) data and transparency, and adopting agricultural subsidies alongside other policy instruments to achieve societal transition. Section 5 presents conclusions.

## 2. Materials and Methods

The purpose of policy instruments is societal change, i.e., technological and behavioural, to achieve policy objectives (see in detail [10]). This article assesses subsidies against legally-binding international environmental goals: Article 2(1) of the Paris Agreement requires limiting global warming to well below 2 °C and pursuing efforts to limit the temperature increase to 1.5 °C above pre-industrial levels. The Aichi Targets B and C of the Convention on Biological Diversity (CBD) require halting biodiversity loss (a post 2020 biodiversity framework is currently being negotiated—see Convention on Biological Diversity [11]; see also goals adopted by the German Federal Constitutional Court (1 BvR 2656/18)). Both agreements imply zero emissions by eliminating fossil fuels in a maximum of two decades and significantly reducing livestock farming, thus calling for comprehensive changes, inter alia, in the agricultural sector [12,13].

Market-based policy instruments such as taxes, cap-and-trade schemes and subsidies increase costs for polluting activities and have been widely discussed as means to support activities which are beneficial for climate and biodiversity, e.g., [14,15]. In earlier publications, we showed that such instruments—especially cap-and-trade schemes—seem to have the largest potential to achieve environmental goals while avoiding typical governance problems such as rebound and shifting effects or enforcement deficits. This is particularly the case if these instruments target major damaging drivers for various environmental challenges such as fossil fuels and if they address an easily graspable governance unit on a broad geographical and substantial scale [12,16–18]. Here we focus on subsidies and aim to assess the extent to which subsidies in general and agricultural subsidies in particular can play a role in transitioning to sustainability.

Given this topic is very broad, we focus on EU agricultural subsidies and underpin the analysis with references to sustainable nutrient (phosphorus) management. We identified the following research questions:

- What are the characteristics of subsidies?

- How do (agricultural) subsidies relate to environmental goals, especially with regard to climate change and biodiversity loss?
- To what extent are (agricultural) subsidies embedded in international environmental agreements (i.e., the Paris Agreement, Convention on Biological Diversity, Sustainable Development Goals)? To what extent does EU law (i.e., CAP and State aid law) promote the application of agricultural subsidies for a transition to sustainability?
- Which (agricultural) subsidies can and should be used to enable a transition to sustainability?

This article applies a qualitative governance analysis including aspects of a legal analysis (Figure 1) (for details [10]). Different researchers understand governance analyses very differently e.g., [19,20]. The qualitative governance analysis applied here aims to identify and assess the effectiveness, i.e., the success in meeting a goal, of existing or proposed policy instruments to achieve normative standards, taking into account knowledge about human motivation and typical governance problems.

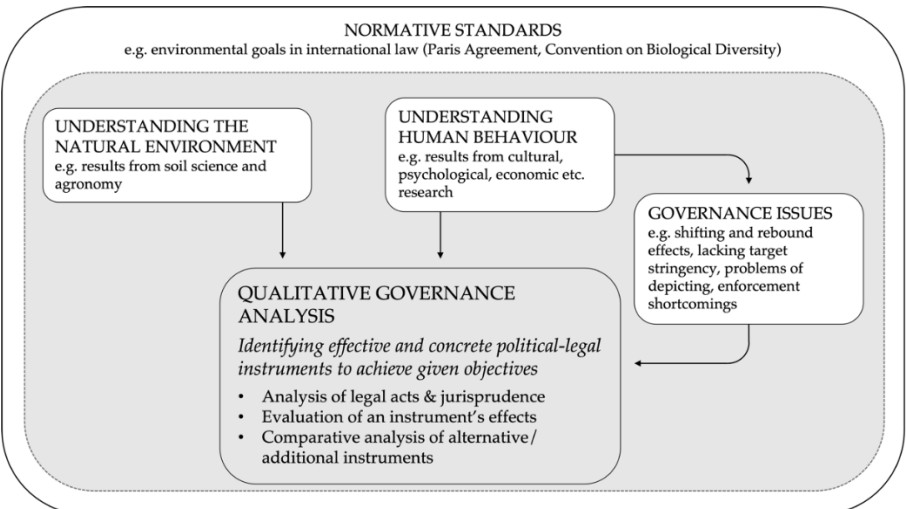

**Figure 1.** Schematic presentation of the qualitative governance analysis (based on [21]).

The qualitative governance analysis builds on natural scientific findings on the environment. In short, it is relevant to the analysis of agricultural subsidies that agriculture performs a dual role with regard to climate and biodiversity: the sector can both harm and benefit climate and biodiversity. Soils play a key role as, for example, well managed soils store substantial amounts of carbon while unsustainably managed soils become a carbon source. Embedded in soil management is nutrient (phosphorus) management, which we discussed extensively in previous publications, e.g., [13,18]. Consequently, when aiming to bring the agricultural sector in line with global environmental goals, it appears sensible to eliminate or reduce harmful activities to the largest extent possible and to promote activities which are beneficial for climate and biodiversity. Nevertheless, substantial uncertainties, e.g., due to the complexity of the natural environment, remain and have to be taken into account in subsequent policy making [22].

Furthermore, the qualitative governance analysis builds on findings of the behavioural sciences. Previous studies on societal change and motivational factors of citizens, consumers, politicians, farmers etc. showed that factors such as self-interest, path dependencies, problems of collective goods, conceptions of normality, and emotional factors such as convenience, denial, habits, scapegoating, and anxiety are important drivers of behavioural change, e.g., [23,24], and are more influential than factual knowledge and values [25,26]. This knowledge provides insights into the effectiveness of existing and proposed policy instruments alongside direct observations of existing instruments, which is important because some governance approaches have never been implemented, and therefore, cannot be observed in reality. Behavioural findings, furthermore, point to potential governance problems of proposed policy instruments (again, besides empirical observations of such



problems). Typical governance problems in sustainability governance include enforcement problems, ambition of targets, rebound effects, and geographical or sectoral shifting effects (e.g., carbon leakage) as well as problems of depicting. While rebound effects describe situations where resource savings or efficiency gains ultimately lead to increasing (or at least not reducing) overall resource consumption, shifting effects describe situations where (environmental) problems are relocated to another place or sector [27–29].

Based on this theoretical background, we performed a literature review. We focussed on studies which discuss agricultural subsidies in the EU and, hence, frequently analyse subsidies of the Common Agricultural Policy (CAP). In short, the CAP is a comprehensive subsidy scheme which is divided into two pillars. The first pillar focusses on income support through hectare-based payments which are conditional upon compliance with the requirements of Conditionality. The second pillar focusses on rural development primarily through, for example, voluntary multi-annual programs such as having perennial wildflower areas. A reformed CAP was adopted by the EU in 2021 (see [30] for details). For our review, we primarily used ScienceDirect, Google Scholar, the reference holdings of the local university library (University of Leipzig) and the databanks of international institutions including the FAO, WTO, OECD and eurostat. The search terms included 'subsidy', 'subsidies', 'agricultural subsidy', 'agriculture' and 'subsidies', 'agricultural incentives', 'economics of subsidies', 'welfare effect subsidies', 'research subsidy environment', 'research development subsidy effect' and 'research development subsidy effect meta'. In addition, the reference lists of the screened literature provided further sources (i.e., a 'snowballing' search method).

This qualitative governance analysis also identifies the normative framework for subsidies. We demonstrate that—alongside governance considerations—legal obligations require more sustainable subsidies. As the analysis focusses on international (environmental) law (i.e., the Paris Agreement, Convention on Biological Diversity, WTO law), we adopt the legal interpretation rules established in the Vienna Convention on the Law of Treaties (Arts. 31 & 32). The methods to interpret legal provisions are: (1) textual, (2) systematic, (3) teleological and (4) historical interpretation. We analyse the legal text itself, consider the broader legal context as well as its purpose and its legislative history, i.e., how a norm has developed [31–33]. Relevant legal documents from the EU were sourced from the official EUR-Lex database (Treaty on the Functioning of the European Union; CAP regulations, State aid law, Green Deal, Farm to Fork Strategy).

Previous studies contributed insights into the discussion on subsidies in general and agricultural subsidies in particular. This article builds on and combines these insights with the legal framework of (agricultural) subsidies.

## 3. Results

### *3.1. Subsidies—Concepts, Strengths and Weaknesses*

There are countless definitions and understandings of subsidies. For example, from an economic perspective, nearly every policy instrument could be defined as a subsidy while legal definitions draw more or less clear boundaries to determine which measures are legally relevant and which are not [34]. Therefore, this section introduces subsidies, and their potentials and limitations as discussed in environmental and ecological economics literature as well as a legal definition. We then analyse the effectiveness of (agricultural) subsidies.

#### 3.1.1. Terminology and Legal Definition

Like other policy instruments, subsidies aim to influence the behaviour of the addressees. In theory, societal change occurs when a subsidy is introduced and makes certain behaviour or certain technological options cheaper and thus more attractive. This can be achieved by direct payments, but also by reducing taxes for certain activities, for example. In turn, eliminating or reducing a subsidy makes certain behaviour more expensive and consequently less attractive. Like other economic instruments, subsidies offer greater freedom to the addressees compared with command-and-control instruments. For example,

while command-and-control instruments prescribe certain behaviours and punish non-compliance, subsidies provide a subject with the choice to accept the subsidy or not [35,36].

Economists use the concept of externalities to make sense of a government's decision to implement subsidies. An externality is a 'cost or benefit arising from any activity which does not accrue to the person or organization carrying out the activity' [37]. Subsidies are used to address positive and negative externalities. This section focusses on subsidies for negative externalities. A subsidy which addresses negative externalities pays polluters for emissions reduction, as Li and Peng [38], and to some degree, Yin et al. [39] and Cao et al. [40] propose. Likewise, the EU provides subsidies to farmers to reduce nutrient inputs into the environment through the establishment of buffer strips [41]. Compared with, for example, a producer tax, a subsidy for emission reduction appears to be an instrument with greater political feasibility [42] but suffers from enforcement problems due to the high number of addressees (i.e., farmers). Moreover, directly subsidising emission reduction can keep emission-intensive producers in production [9] and induce additional market entry if entry is sufficiently easy. Hence, these subsidies cause rebound effects [43] (Section 2). Lastly, the provision of subsidies to correct negative externalities contradicts with the polluter pays principle which is established in EU primary law (Article 191 (2) TFEU). Against this background, it appears that using subsidies to correct negative externalities is ineffective and not in line with EU primary law. Moreover, where subsidies support environmentally harmful activities such as fossil fuel use, they directly promote one central driver of climate change and biodiversity loss which contradicts the goals of the Paris Agreement and Convention on Biological Diversity. This aspect is very important since the overall amount of environmentally harmful subsidies is substantial, as we show in the following section (see also Section 1).

Economists have also sought to identify the characteristics of (formal) subsidies which include: (1) the provider, (2) the type of support, (3) the addressee, (4) the conditions attached and (5) the consignee. Usually, subsidies are provided by a public authority (the provider) as direct or indirect subsidy (type of support). While direct subsidies involve actual payments such as the direct income support of the CAP, indirect subsidies do not involve actual payments. Instead, a public authority forgoes income while the beneficiary receives a benefit through a tax concession, for example. Addressees may be public or private, businesses or service providers. Subsidies do not require a financial return service from the beneficiaries. However, direct or indirect support may be conditional on compliance with legal requirements (see number 4 above). An example is income support of the CAP linked to compliance with Conditionality which e.g., requires farmers to establish buffer strips along water courses to reduce phosphorus runoff. The extent of conditions attached to a subsidy affects the freedom of the subsidy receiver [36,44,45]. Figure 2 provides an overview of the systematisation of subsidies.

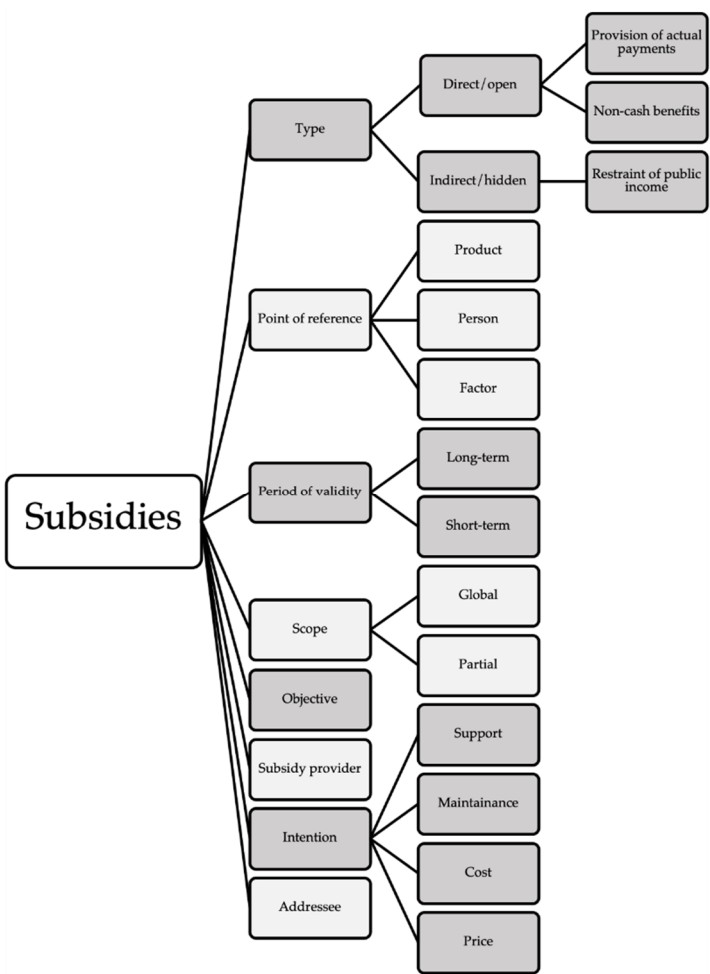

**Figure 2.** Systematisation of subsidies. Own figure based on [44] (pp. 23–31), [45] (pp. 275–279).

While formal subsidies affect the behaviour of the subsidy receiver through government action, informal subsidies affect behaviour through government inaction. Thus, some have categorised negative environmental externalities as an informal subsidy [46–48]. Accordingly, where a farmer excessively uses mineral fertilizers to grow crops, declining groundwater quality and the eutrophication of water bodies would qualify as informal subsidy because the farmer avoids the cost of cleaning up and gains a benefit [49]. Others exclude the lacking internalisation of external effects from their analysis. They argue that this problem reflects a general issue of environmental policy making rather than being a result of providing (indirect) benefits to some. However, this approach reduces the precision of subsidy estimates [50,51]. Besides negative environmental externalities, some identify the non-taxation of, e.g., kerosene use, as a subsidy. Therefore, not only is an assessment of tax incentives necessary, but also the scope of a tax program [50,52]. Less visible formal subsidies include differing environmental protection measures between countries (called regulatory subsidies by [53]), and the provision of infrastructure [54,55].

There appears to be little disagreement that less visible subsidies are a substantial issue. Yet, their identification and quantification are challenging. Investigating these subsidies is useful for making environmental damage and free riding visible. In doing so, these investigations can engage the public and pressure political decision-makers to effectively address environmental issues. Therefore, precise terminology and collecting complete and reliable data are important. The problem of less visible subsidies further illustrates that climate and environmental protection have to be consistently integrated into all policy areas to ensure that the goals of the Paris Agreement and the Convention on Biological Diversity are achieved.

As indicated above, legal definitions determine which measures are relevant for the law and which are not. This section introduces the definition of State aid, which serves as a basis for Section 4.2. One objective of the EU is to establish a single market with little competitive distortion. The Treaty on the Functioning of the European Union (TFEU) contains provisions on 'State aid' rather than 'subsidies'. State aid law addresses trade between Member States (i.e., intra-union trade). State aid law is part of the competition policies. Any State aid that (potentially) negatively affects the internal market of the EU, i.e., the competition between the Member States, is principally prohibited. The Commission functions as an administrative supervisory body. Early on, the European Court of Justice clarified that the concept of Aid is wider than that of a subsidy as subsidies only provide positive benefits [56]. In doing so, the Court takes a very narrow understanding as to what constitutes a subsidy. In 1988, the Commission followed this approach: State aid is an instrument to pursue the objectives of the EU [57]. Likewise, the glossary of the Commission defines State aid as being delivered in different ways, including the allocation of grant subsidies [58].

Article 107(1) of the TFEU establishes the characteristics of State aid. State aid exists if (1) aid is granted by a Member State or through the resources of a Member State, (2) it (is likely to) negatively affect the competition between Member States and (3) it transfers a benefit (4) to certain activities or the production of certain goods (selectivity), and (5) it affects trade between the Member States [59,60]. The form of the State aid is not a relevant criterion, neither are its objective or causes. Instead, its effect is the defining element (Figure 1) [61,62] (para 17). While subsidies under WTO law have to be specific and State aid selective, in contrast to WTO law, State aid has to have a burden on the state's budget. The Commission clarifies that a 'positive transfer of funds does not have to occur; foregoing State revenue is sufficient. Waiving revenue which would otherwise have been paid to the State constitutes a transfer of State resources' [63] (para 51). There is no separate definition for agricultural State aid. Still, there are specific rule for State aid in the agricultural sector which will be analysed further below. Overall, subsidies aim to change behaviour through pricing, they take multiple forms, and can be more or less visible, which makes their assessment challenging.

3.1.2. Potentials and Limitations of Subsidies in General

Subsidies can be adopted for different reasons. This section analyses subsidies in general as well as the provision of public goods (or positive externalities), including research and development (R&D), as they are particularly relevant for agriculture.

Subsidies can be a tool to achieve universal or more equal access to consumption or income. Typical examples include food subsidisation and fuel support [64]. The State aid law of the EU establishes that subsidies having a social character and granted to individual consumers are permitted in the internal market of the EU (Article 107(2)(a) TFEU). While theory suggests that targeted transfers are the more effective redistributive tool, assessing who qualifies for the transfers and transferring funds is frequently associated with large transaction costs. This makes undiscriminating consumption subsidies the easier alternative [65]. Furthermore, to achieve global environmental goals, it is more effective to combine direct social transfers with ambitious environmental policy instruments rather than aiming at social goals by less ambitious—and therefore, at first glance less costly—environmental instruments [10,66]. Yet, these subsidies may have negative distributional consequences (see below). In addition, subsidies can quickly make funding available [67], such as for support programs in response to natural disasters.

Another textbook justification for using subsidies is the provision of public goods. Producers provide products or services, but the market price does not reflect their value. The market price may be too low, or a market may be absent [36,67,68]. The agricultural sector is closely related to several public goods and can simultaneously provide public goods such as the enhancement of biodiversity and increases in the carbon storage potential of soils (Section 2). Therefore, using subsidies to provide public goods appears to be an

effective approach to combatting global environmental challenges including climate change and biodiversity loss.

Subsidies are implemented for R&D. Studies have investigated the relationship between R&D and reductions in greenhouse gas GHG emissions (for an overview see Long et al. [69]) by measuring the relationship between innovative activity and emission intensity rather than absolute GHG emissions. While some find that emission intensity is negatively correlated with research in low emission technology [70–72], others show that research succeeded in improving energy efficiency but failed to reduce the carbon intensity of energy production itself [73]. However, only considering the impact on GHG intensity likely overstates the contribution of innovation towards decarbonisation. This is because of rebound and scale effects, e.g., [74]. The scale effect describes the positive correlation between R&D and GDP growth, which typically raises energy demand and GHG emissions. Therefore, others investigated the effect of different innovation variables on aggregate GHG emissions. While Li & Wang [74] found a net negative relationship between aggregate R&D investment and aggregate emissions, others found mixed results, though their evidence shows a net emissions reduction with increased research efforts for most cases [75–78]. Other researchers have modelled the effects of a carbon price, R&D subsidies, or a combination of both policies and found that an optimal policy mix would initially include both as inducing the required technical change through a carbon price alone would lead to an excessive loss in income ([79,80], supported by [81–84]). In summary, there appears to be a strong case for the use of R&D subsidies in transitioning to a net-zero GHG economy. Spending reduces GHG emissions, particularly when targeted at low- and no-emission technologies [74], yet typical governance issues frequently appear.

Research on the aggregate effect of R&D on GHG emissions in the agricultural sector is sparse. Long et al. [69] show that patent applications in the agricultural sector in China are negatively correlated to the country's agricultural $CO_2$ emissions (while omitting other GHGs). Still, there is no apparent reason why the relationship of GHG emissions and innovation should be fundamentally different in the agricultural sector than in the rest of the economy. Despite this, public agricultural R&D investment in high income countries has slowed down in recent years [85]. Thus, increasing subsidies for research on mitigation technologies such as manure and slurry storage and processing, and optimal fertiliser application appears effective. It should be noted though that already developed technologies with a mitigation potential are not always readily adopted in agricultural production [86–91]. There are several barriers slowing down adoption (see [89] for an overview), including economic viability. Reducing these barriers is essential.

In public discussion, subsidies frequently appear to be a superior policy instrument since they supposedly hurt no one, although they are paid by taxpayers. Yet, subsidies have shortcomings. Here, we focus on their (1) costliness, (2) distributional issues, and (3) removal problems.

(1) Subsidies are expensive and directly and indirectly pressure public finances [68,92–94]. For instance, in 2015, global subsidies for coal (3.9%) and oil (1.8%) amounted to almost 6% of global GDP [95]. In the EU, the CAP swallows nearly one third of the Multiannual Financial Framework between 2021 and 2027 [96]. On a national level, according to the German Federal Ministry of Finance, federal subsidies through direct financial aid and tax credits for 2020 were (before the pandemic) estimated at roughly EUR 31.5 billion. This accounts for about 8.7% of the planned budget [97].

(2) Subsidies have distributional consequences. When aimed at universal access, subsidies are frequently ill targeted and have adverse distributional effects [94,98]. For example, relative to their income, the less well-off often profit more from a subsidy, but in absolute terms, the wealthier receive larger benefits due to higher consumption levels. As a comprehensive subsidy scheme, CAP subsidies also have adverse distributional effects. Most full-time farm employees work in the poorest regions of the EU (measured by average farm-labour income) while the fewest full-time farm employees are in the regions with the highest farm-labour income. Yet, the regions

with highest farm-labour income receive comparatively more CAP subsidies. In contributing to income inequality, the CAP runs counter to its policy objective, i.e., to ensure a fair standard of living [99,100] (Article 39(1) (b) TFEU). Moreover, the agricultural subsidies of the EU are likely to affect third-world countries. For instance, the direct payments and export subsidisations of the past have pushed local food producers in third-world countries out of the market and hampered the development of independent local food industries, particularly in Africa [101,102]. While recent research is unable to paint a clear picture of the effects of current agricultural subsidies on developing countries [103], NGOs claim that CAP subsidies continue to hamper the establishment of local production in developing countries to the present [104,105].

(3) Once enacted, it can be very challenging to remove a subsidy. The reasons exemplify the motivational factors of different actors (Section 2). Oftentimes, the self-interests of interest groups oppose reforms and successfully influence the policy process. For example, the influence of the agricultural lobby of the EU is substantial [106] (for the influence on energy policy see [107]). The interest group of European farmers COPA-COGECA has exclusive access to informal meetings with agricultural ministers during trilogue negotiations of the CAP [108]. Another aspect of self-interest is that policy makers worry about alienating their voting base when removing subsidies [46,109,110]. In addition, individuals' loss aversion and tendencies for simplification are relevant (Section 2). The cost of a particular subsidy is hardly discernible and oftentimes not particularly large for an individual. Furthermore, collective good problems arise. There is commonly little incentive for individuals to engage in collective action, even though the overall societal cost of a subsidy may be very large. In turn, some individuals usually profit substantially from certain subsidies so aim to have them continued [111] (for discussion on past CAP reforms see [112]).

In summary, there appear to be legitimate policy objectives to use subsidies, especially for social policy, but also for providing public goods. The agricultural sector offers several of these policy objectives given its dual role regarding climate change and biodiversity loss (Section 2). Yet, subsidies frequently have shortcomings. It has, therefore, been proposed to apply subsidies in complementary roles [113,114]: Subsidies should only be implemented for targeted activities rather than as broad instrument. This not only requires eliminating environmentally harmful subsidies but also reassessing the prominent role of subsidies in agricultural policy in general. The following section analyses subsidies in the agricultural sector.

### 3.1.3. Agricultural Subsidies

This section analyses the special features of the agricultural sector and introduces the three subsidy types which are used in the EU (coupled, decoupled and agri-environment-climate commitments). The agricultural sector offers some peculiarities regarding subsides given:

1. The large volume of subsidies (Section 1),
2. The dual role of agriculture regarding climate and biodiversity (Section 2),
3. The heterogenous policy objectives which can result in contradictory incentives ([106,115,116]; Article 39 TFEU),
4. The substantial influence of interest groups [93,117,118] (Section 3.1.2),
5. The likelihood of spillovers, e.g., unsustainable fertilising practices on agricultural land can negatively affect water bodies through nutrient leaching [118,119].

Agricultural subsidies have different reference points, e.g., input and production (Section 3.1.1), and target different levels. Farm level subsidies are divided into coupled and decoupled subsidies. Coupled subsidies are linked to, e.g., production levels while decoupled subsidies are not. The direct payments of the first pillar of the CAP include coupled and decoupled subsidies. Approximately 90% of the budget for direct payments is allocated to decoupled subsidies, and 10% of this budget is allocated for coupled subsidies [120]. Coupled and decoupled subsidies affect the environment differently. Coupled subsidies are frequently environmentally harmful because they, for example, incentivise

high inputs of mineral fertilisers [6,28,121], although more research on the effects of coupled subsidies is needed. Besides, three quarters of coupled support in the EU is provided for the livestock sector [120], which is generally associated with large resource consumption. Decoupled subsidies appear to be the least environmentally harmful agricultural subsidies [122,123]. The environmental effectiveness of decoupled subsidies depends on the conditions attached to these subsidies [124]. The past conditions of decoupled CAP subsidies, i.e., cross-compliance and greening, have been judged as environmentally ineffective, e.g., [125–127]. Overall, the CAP is associated with multiple environmental issues, e.g., [128,129], which shows that these least environmentally harmful decoupled subsidies are not in line with global environmental goals. (Decoupled) subsidies fail to address the drivers of climate change and biodiversity loss, e.g., fossil fuels and livestock farming. Instead, these subsidies frequently even push these drivers. Moreover, all farm level subsidies suffer from enforcement problems due to the high number of addressees.

The subsidies of the second pillar of the CAP are partly provided for the provision of public goods. The Member States of the EU have to provide subsidies for agri-environment-climate commitments. Most subsidies are provided as action-based subsidies, i.e., for predefined actions that are expected to achieve certain environmental outcomes. Yet, Member States may also provide results-based subsidies which are conditional upon achieving environmental outcomes rather than actions performed. Many studies have investigated the effectiveness of these agri-environment-climate commitments to, for example, enhancing biodiversity. Some studies find that, in general, agri-environment-climate commitments have been beneficial for farmland biodiversity [106,130–132]. Others find mixed [133] or limited effects [134]. To enhance their effectiveness, these subsidies require quantifiable and differentiated objectives [133]. Depending on these objectives, subsidies which, for example, target species with specific requirements will not only have to consider local environmental factors but also the requirements of the species. Other subsidies may, in turn, be effective when prescribing general obligations [132,133,135,136]. Moreover, subsidies seem effective when targeting intensively farmed farmland [130,132], for example, where water bodies are at high risk from diffuse agricultural pollution [134] such as phosphorus runoff. Adopting a long-term perspective appears useful [137,138] as some biodiversity effects only occur after multiple years [135]. Lastly, studies highlight the importance of adapting subsidies to local circumstances [128,130,133,134,136,137,139,140]. However, if subsidies turn into small-scale instruments, they increase the administrative burden and are likely to incentivise shifting effects where unsustainable agricultural practices might be relocated to other areas. Collaborative landscape subsidies might be able to limit this governance problem to some extent.

### 3.2. Legal Obligations towards More Sustainable Subsidies

This section demonstrates that—besides governance insights—legal obligations also require more sustainable subsidies.

### 3.2.1. Subsidies in Transnational Environmental (Soft) Law

It might appear odd to screen international environmental law and soft law for a specific policy instrument. Still, this section investigates subsidies in non-legally binding provisions such as the Sustainable Development Goals (SDGs) as well as in legally binding provisions such as the Paris Agreement (PA). This is because international environmental (soft) law is—in contrast to other policy instruments—frequently quite specific about the abolishment of environmentally harmful subsidies (see [141] for a past review). One reason for this might be the enormous scale of environmentally harmful subsidies (Section 3.1.2) and the repeated political statements to reform and phase out these subsidies (Section 1). At the same time, there appears to be political consensus that enabling a transition to sustainability requires substantial financial resources (see below).

International environmental (soft) law frequently establishes explicit references to the abolishment of environmentally harmful subsidies. However, there is less clarity about

keeping or introducing environmentally beneficial subsidies. This is partly due to imprecise semantics: the word subsidy mainly occurs in the context of abolishing environmentally harmful subsidies while words such as 'incentives', 'support', and 'financial resources' are used to argue for the introduction of environmentally beneficial subsidies. The sustainable development goals (SDGs) serve as example. SDG 12.C requires Parties to rationalise inefficient fossil-fuel subsidies by phasing out harmful subsidies. SDG 2 aims for zero hunger by, for example, promoting sustainable agriculture. To this end, the Parties to the Agreement are to eliminate 'all forms of agricultural export subsidies' (SGD 2.B). The two subgoals are explicit about the abolishment of subsidies. In contrast, SDG 15.A requires the Parties to mobilise and increase 'financial resources' to conserve and sustainably use biodiversity and ecosystems. Whether this includes mobilising private capital or implementing subsidies or both remains unclear. Formulating more precise documents would be useful to support policy makers in their decision making and address the dual role of subsidies, i.e., eliminating harmful subsidies and developing environmentally-beneficial subsidies to achieve a transition to sustainability.

The Paris Agreement

This section analyses two provisions of the Paris Agreement. While other provisions also relate to subsidies, for reasons of space, this section focuses on Article 2(1)(c) PA and Article 9(3) PA.

At first sight, the Paris Agreement appears vague on subsidies. The seldomly discussed Article 2(1)(c) PA requires 'making finance flows consistent with a pathway towards low greenhouse gas emissions and climate-resilient development'. There is ambiguity about the scope of finance flows. For example, do finance flows include both, public and private finance flows (acknowledging that both are interrelated)? An inclusion of public finance flows would imply that Article 2(1)(c) PA covers subsidies (as discussed in Section 3.1.1, subsidies include a financial contribution from a public authority). We support the argument of others that the provision indeed includes both, public and private finance flows [142–144].

The argument is based on a systematic interpretation of Article 2 PA in light of Article 9(3) PA. Article 9 PA covers international assistance from developed to developing countries. Article 9(3) PA requires developed country Parties to mobilise climate finance from a wide variety of sources, instruments and channels, noting the significant role of public funds. Consequently, the provision incorporates non-public (i.e., private) funding and public funding. Article 3 PA links the provision of Article 9 to Article 2 PA by requiring that the efforts undertaken in international assistance are to achieve the purpose of the Agreement established in Article 2 (see also [145], para 52). Hence, international assistance through climate finance appears to be a policy instrument which, among other objectives, aims to achieve sustainable finance flows. Given Article 9(3) PA includes public and non-public sources, this scope transposes to Article 2(1)(c) PA.

A historic perspective supports the coverage of subsidies in Article 2(1)(c) PA. Article 2(1)(c) PA emerged from a textual proposal which required the Parties to 'reduce international support for high-emission [and maladaptive] investments and enhance international support for low-emission and climate-resilient investments' [146]. The Parties to the Agreement, which are governments only, have to take action. In spite of this, a recent conclusion of EU finance ministers states that phasing out environmentally harmful fossil fuel subsidies is merely a 'key component of an enabling environment' to shift financial flows [147]. In contrast, a proposal for a WTO Ministerial Statement highlights the direct potential of a fossil fuel subsidy phase out to achieve Article 2(1)(c) PA [3] (para 5). In summary, Article 2(1)(c) PA covers subsidies.

Furthermore, the question arises how Article 2(1)(c) PA relates to the temperature objective of the Paris Agreement. Article 2(1) PA lists three objectives. Firstly, the Agreement requires that global warming is kept well below 2 °C while pursuing efforts to 1.5 °C above pre-industrial levels (mitigation). Secondly, the Agreement requires adaptation, and thirdly,

it requires that finance flows are made consistent with a pathway towards low greenhouse gas emissions and climate-resilient development. The Paris Agreement builds on the United Nations Framework Convention on Climate Change (UNFCCC). The goal of the UNFCCC is to achieve a stabilisation of greenhouse gas concentrations in the atmosphere at a level that would prevent dangerous anthropogenic interference with the climate system (Article 2 UNFCCC). Interpreting the provisions of the Paris Agreement in light of other environmental treaties such as the UNFCCC (Article 31(3) Vienna Convention) implies that the mitigation goal of Article 2(1)(a) PA is superordinate to other norms of the Agreement [148].

Article 2(1)(c) PA could be interpreted as a means to achieve the mitigation and the adaptation goal because 'a pathway towards low greenhouse gas emissions' refers to mitigation and thus to Article 2(1)(a) PA, while 'climate-resilient development' refers to adaptation and thus Article 2(1)(b) PA (see also [146]). At the same time, Article 3 PA mentions Article 2 PA as the 'purpose' of the Agreement. This suggests that Article 2(1)(c) PA is a goal itself. We conclude that making finance flows consistent with a pathway towards low greenhouse gas emissions and climate-resilient development is a means *and* a goal of the Paris Agreement (see also [142], interpretation disputed in [146]). Subsidies, being one element of 'finance flows', will therefore, have to be consistent with a pathway towards low greenhouse gas emissions and climate-resilient development, and contribute to mitigation, i.e., the temperature goal of the Agreement.

Lastly, finance flows (Article 2(1)(c) PA) differ from climate finance (Article 9(3) PA). While climate finance covers measures with a positive contribution to climate change, finance flows are to be made 'consistent' towards low greenhouse gas emissions and climate-resilient development. This encompasses adopting finance flows which positively contribute to climate change and eliminating finance flows which hinder combating climate change. Consequently, while Article 2(1)(c) PA does not explicitly refer to eliminating environmentally harmful subsidies, subsidies which run counter to a pathway towards low greenhouse gas emissions and climate-resilient development, are nevertheless, not consistent with the provisions of the Paris Agreement [110]. The agricultural sector fits well into this frame due to its dual role regarding the environment (Section 2). Overall, eliminating subsidies which support harmful activities and instead implementing targeted subsidies which enhance environmentally beneficial farming practices is essential (Section 3.1.3). This would make finance flows to the agricultural sector consistent towards low greenhouse gas emissions and climate-resilient development.

Convention on Biological Diversity

Even though the parties to the Convention on Biological Diversity are currently negotiating a post-2020 global biodiversity framework, this section analyses one Aichi Target of the Strategic Plan for Biodiversity 2011–2020 as these are the most relevant provisions on biodiversity (Table 1).

**Table 1.** Strategic Goal A of the Convention on Biological Diversity.

| Strategic Plan for Biodiversity 2011–2020 | |
|---|---|
| Strategic Goal A | Address the underlying causes of biodiversity loss by mainstreaming biodiversity across government and society |
| Target 3 | By 2020, at the latest, incentives, including subsidies, harmful to biodiversity are eliminated, phased out or reformed [ . . . ], and positive incentives for the conservation and sustainable use of biodiversity are developed and applied [ . . . ]. |

While it is positive that Target 3 establishes an explicit reference to the abolishment of environmentally harmful subsidies by 2020, the Conference of the Parties in 2018 highlighted the lack of progress on the elimination, phase out, or reform of incentives, including

subsidies, that are harmful for biodiversity [149] (para 9). This exemplifies the challenge to reform environmentally harmful subsidies (Section 3.1.3). Besides, given that the Strategic Plan lacks definitions, the scope of subsidies remains unclear. Therefore, for the current negotiations, it would be useful to close this gap.

On a superordinate level, the Convention on Biological Diversity frequently points to subsidies. This section focuses on Articles 11 and 20. Article 11 requires the Parties to, as far as possible and as appropriate, adopt measures that act as incentives for the conservation and sustainable use of components of biological diversity. While the discussion above concludes that providing subsidies for public goods such as the conservation of biological diversity in agriculture is useful (Section 3.1.3), Article 11 is very weak due to the qualifier 'as far as possible and as appropriate' [150]. Likewise, Article 20(1) establishes that a party shall aim to provide, within its capabilities, financial support and incentives for activities which are intended to achieve the objectives of the Convention, in accordance with its national plans, priorities and programmes. The provision provides much flexibility by emphasising that financial support and incentives need to be embedded into national contexts. Interestingly, Article 20(1) differentiates between financial support and incentives without specifying their characteristics. Apart from that, like the climate finance of the Paris Agreement, both Articles discuss subsidies which are beneficial for achieving the goals of the Convention—rather than requiring the abolishment of environmentally harmful subsidies—and thereby miss an important element.

### 3.2.2. Subsidies in EU (Soft) Law

This section analyses subsidies in the EU Green Deal and the Farm to Fork Strategy, both legally non-binding strategies.

### Green Deal and Farm to Fork Strategy

The European Green Deal is a framework for multiple policy initiatives, including the Farm to Fork Strategy, which collectively aim to make the EU economy sustainable [151]. In the Green Deal, the Commission aims to end fossil fuel subsidies and revise current tax exemptions (p. 10). National budgets are to become green based on a redirection of public investments away from harmful subsidies alongside tax reforms. These tax reforms are intended to send 'the right price signals and [provide] the right incentives for sustainable behaviour' (p. 17) and thereby address environmentally harmful behaviour and environmentally beneficial behaviour. On an international level, the EU aims to enhance cooperation which may include ending global fossil fuel subsidies (p. 21). While it is positive that environmentally harmful subsidies have been implemented in the Green Deal, the proposed measures cover only certain sectors (i.e., mobility) which could lead to shifting effects, and it is unclear if, for example, diesel tax exceptions for farmers are covered in the mobility sector. Lastly, the Green Deal refers to the CAP as a key instrument to combating climate change and biodiversity loss in agriculture. Yet, rather than contributing to combat these challenges, the subsidies of the CAP contribute to multiple environmental problems such as degraded soils and eutrophication [30,152].

The Farm to Fork Strategy sits between the Green Deal and the CAP. It was published in 2020 and aims to make the food system of the EU sustainable [153]. To ensure that all food becomes sustainable, the Commission will publish a legislative proposal for a framework for a sustainable food system. Alongside common definitions, certification and labelling, the framework will also include 'targeted incentives' to enhance sustainability standards (pp. 5–6). It is unclear if the Commission proposes a new incentive mechanism for the agricultural sector parallel to the CAP, or if 'targeted incentives' will be included in the CAP. In addition, the Farm to Fork Strategy discusses tax incentives to enable a transition towards sustainable food systems and healthy diets. EU tax systems should ensure that the price of different foods reflects their real costs in terms of use of finite natural resources, pollution, GHG emissions and other environmental externalities (p. 14). The Farm to Fork Strategy also emphasises research and innovation. Under Horizon Europe, the Commission

is developing a mission on soil health and food. The mission aims to develop solutions for restoring soil health and functions which would contribute to reducing external inputs into soils such as fertilisers (pp. 15–16). Overall, as non-binding instruments, the effects of the Green Deal and the Farm to Fork Strategy will be limited. The limited effectiveness is also highlighted by the Commission's omission of the Green Deal and the Farm to Fork Strategy's objectives when assessing the CAP Strategic Plans of Member States [154].

EU State Aid and Agriculture

Member States of the EU can adopt domestic agricultural subsidies outside the CAP if they comply with State aid rules. State aid plays a minor role in the agricultural sector because most of the agricultural subsidies of the EU are provided through the CAP rather than domestic measures, which are covered by Article 107–109 TFEU. Still, as State aid for agriculture is seldom analysed, this section examines the topic. If a Member State intends to implement or alter State aid, it has to notify the Commission. The Commission will assess if the State aid is compatible with the internal market (ex ante assessment). When the Commission reaches its decision, the Member State can adopt the measures (Article 108(3) TFEU). In general, State aid which distorts or threatens to distort the market of the EU is prohibited. Yet, it is assumed that certain State aid is necessary 'to address market failures in order to ensure a well-functioning and equitable economy' [155] (Part I, Chapter 1(2)). Exempt from the principal prohibition and with regard to the agricultural sector is State aid to counteract damages by natural disasters (Article 107(2) TFEU). In addition, the Council can define State aid categories which are exempted from the notification requirement (Article 109 TFEU). The Council may also—if justified by exceptional circumstances—declare any other State aid by Member States to be compatible with the internal market (Article 108(2) TFEU). As such, the legislator has much leeway and flexibility in approving the State aid of Member States, yet excludes the European Parliament from co-decision making [156].

The General Block Exemption Regulation (651/2014) establishes that aid granted for environmental protection does not need to be notified to the Commission; however, the Commission is to be informed about the measures. In fact, 95% of State aid measures are covered by this regulation and thus do not need to be notified and approved by the Commission [157]. De minimis aid is governed in a separate regulation (Regulation (EU) 1407/2013). De minimis aid shall be declared compatible with the internal market and does not require notification and information to the Commission if the aid granted per Member State to a single undertaking does not exceed EUR 200,000 over three years (Article 3(2) Regulation (EU) 1407/2013). It is assumed that aid granted under this ceiling does not affect trade and competition between the Member States (Regulation (EU) 1407/2013, p. 1). Exempted from these provisions is aid granted to the production of agricultural products and certain activities in the marketing and processing of these products (Article 1(1) Regulation (EU) 1407/2013).

The CAP Strategic Plan Regulation contains the rules that determine which support is subject to competition rules (Article 107–109 TFEU). Principally, the rules on competition apply to CAP support. However, exemptions include payments made by Member States within the context of that regulation [CAP SP Regulation] as well as additional national financing which falls within the scope of Article 42 TFEU (Article 145(1), (2) CAP SP Regulation). Consequently, the majority of CAP subsidies lies outside State aid provisions as direct payments only account for three quarters of the CAP budget (Section 3.1.3). For the remaining subsidies, i.e., those to which State aid rules principally apply, the Commission has adopted additional measures: (1) certain aid is exempted from the notification requirement, (2) there are specific rules for de minimis aid, and (3) there are guidelines for State aid in the agricultural sector. This section focuses on measure (1).

The EU provides rules for highly detailed aid constellations which do not require notification. Instead, information has to be provided to the Commission (block exemptions), upon the condition that the aid is provided to small and medium-sized enterprises (ABER Regulation (EU) 702/2014). The ABER provisions act as framework through which Member

States can implement State aid. The regulation includes six categories of State aid. The regulation establishes that aid for investments into assets on agricultural holdings linked to primary agricultural production per undertaking per investment object up to EUR 500,000 do not require notification to the Commission (Article 4(1)(a) Regulation (EU) 702/2014). These investments must pursue at least one of the objectives listed in the regulation. Although the objectives include environmental and climate objectives, the investments may also be undertaken to create and improve the infrastructure related to the development, adaptation and modernisation of agriculture, such as access to farm land (Article 14(3) Regulation (EU) 702/2014). Hence, these subsidies are not always contingent upon environmental objectives. Besides, investment support is primarily accessed by large farms and contributes to the concentration of land [158], which enhances the negative distributional effects of CAP subsidies (Section 3.1.2). It would be useful to make all agricultural State aid contingent upon environmental objectives to ensure subsidies are used for transitioning the agricultural sector to sustainability.

The Commission has published guidelines for State aid in the agricultural sector which it uses to assess proposed State aid which is not State aid for small and medium-sized enterprises or de minimis aid and, therefore, exempted from notification [155]. Where State aid is in line with the guidelines, the Commission assumes it is principally compatible with the internal market. State aid can be provided to small and large enterprises and includes, for example, nationally funded rural development schemes outside the CAP, and aid for livestock [155] (23). The Commission assesses if 'the positive impact of the aid towards an objective of common interest exceeds its potential negative effects on trade and competition' [155] (38). State aid has to satisfy seven criteria, including its contribution to a well-defined objective of common interest, appropriateness of the measure, having an incentive effect and being transparent [155] (39). Regarding the contribution to a common objective, the guidelines establish that aid in the agricultural sector has to ensure viable food production and promote efficient and sustainable resources use to achieve intelligent and sustainable growth [155] (43). In addition, the guidelines refer to Article 11 TFEU which covers 'environmental protection' and the polluter pays principle. Yet, the fact that agricultural subsidies—unless provided for environmental protection and other public goods—generally contradict the polluter pays principle is not discussed. Furthermore, the guidelines are highly complex and require substantial resources in public administration [159]. For example, the guideline defines 37 different aid categories for the agricultural sector. Reducing the complexity could free resources that could be used to design targeted (agricultural) subsidies.

To counteract the insufficient alignment of State aid provisions in the agricultural sector, the assessment criteria of the Commission could be amended for example, by incorporating a mandatory assessment towards global environmental goals. The Commission is currently revising the ABER Regulation to better promote aid for implementing the Green Deal [160], which is welcome but will most likely be insufficient. Moreover, given that most State aid does not require approval by the Commission, amended assessment criteria are not sufficient. Instead, the requirement for an alignment with global environmental goals has to trickle down into all guidelines.

Lastly, State aid decisions of the Commission are subject to judicial review by the European Court of Justice. The Treaty provides that any natural or legal person can institute proceedings against an act—such as a State aid decision—which directly and individually affects them [61] (Article 263 TFEU). However, NGOs have never been admitted in State aid cases, which is highly problematic [161]. This issue has been investigated by the Aarhus Convention Compliance Committee. The Aarhus Convention requires that its signatories including the EU provide members of the public with access to judicial reviews so that they can challenge acts which contravene environmental law (Article 9(3) Aarhus Convention). The Committee determined that the EU fails to comply with this provision (ACCC/C/2015/128). The EU will have to implement the ruling in policy and recently announced a political agreement [162]. Thus, this ruling is expected to translate into greater

transparency and public scrutiny of State aid decisions and offers the possibility to pressure policy makers to better align State aid decisions with global environmental goals.

## 4. Discussion: The Way Forward for Subsidies

### 4.1. Reform and Removal of (Agricultural) Subsidies

Having analysed the effectiveness of subsidies with regard to global environmental goals as well as the legal framework of subsidies, this section discusses the consequences of these results.

Removing harmful subsidies is challenging and the environmental effects of removing harmful agricultural subsidies are limited measured against global environmental goals [8,28,163,164], although this is contested by [165]. This raises the question why there is an urgency to remove environmentally harmful subsidies. Firstly, the studies cited above focus on GHGs only. They do not factor in other effects such as impacts on distribution and biodiversity, and as a result, most likely underestimate the overall impact of subsidy removal. Secondly, (agricultural) subsidies are covered by Article 2(1)(c) of the legally binding Paris Agreement which calls to make finance flows consistent with a pathway towards low greenhouse gas emissions and climate-resilient development (Section 3.2.1). Removing subsidies which are not in line with the goals of the Paris Agreement and the Convention on Biological Diversity and also comprehensive subsidy reforms are needed. Past subsidy reforms were not effective: 'Making agricultural subsidies conditional on use of lower-emission approaches is a tempting approach but appears to have had relatively little impact' [8]. Likewise, CAP reforms indicate that decoupling agricultural subsidies and making them conditional on environmental requirements does not bring them in line with global environmental goals (see Section 3.1.3). Therefore, reforming subsidies must to go further by, for example, removing damaging income support subsidies, designing targeted subsidies and increasing funding for R&D. However, subsidy removal requires careful policy making to avoid typical governance issues (see Section 2). For example, removing coupled subsidies is likely to lead to emission leakage, i.e., shifting production to third countries [28]. The Carbon Border Adjustment Mechanism that was recently proposed by the Commission could counteract these effects.

A comprehensive reform of agricultural subsidies in the EU also requires reassessing the CAP objectives, which are established in the Treaty on the Functioning of the EU (Section 3.1.3), because:

1. Agricultural subsidies frequently produce ambivalent effects regarding these objectives. Nutrient management of the Baltic Sea exemplifies how policy objectives are in conflict with each other. Achieving effective nutrient reductions, i.e., the policy objective of environmental protection, requires substantial costs and restrictions on agricultural practices. This will likely increase food prices, which runs counter to the policy objective to provide affordable food [166].

2. Agricultural subsidies frequently support practices which contribute to climate change and biodiversity loss (Section 3.1.3). The CAP objectives manifest the status quo and few CAP objectives have a long-term perspective. Instead, short-term perspectives are dominant [167]. In doing so, the CAP core principles stand in the way of transitioning the agricultural sector to be in line with global environmental goals.

3. Even though income support holds a high priority in agricultural policy in the EU, farm income support is just one among multiple policy objectives. The OECD notes that countries around the globe vary considerably in the extent to which they consider income support as a policy objective [6]. Although there may be benefits in farm maintenance, earning an adequate income is not a public good [14] (pp. 37–38) [168]. It is, therefore, questionable whether the majority of agricultural subsidies should be allocated to this objective. Moreover, the Member States of the EU have social security systems, so if their income is too low, farmers can—like other citizens—request support through the general social security systems (Section 3.1.2).

Having said that, the CAP is more than a subsidy scheme. For example, the CAP includes common rules on competition within agricultural markets (Article 40(1) (a) TFEU). While for these rules too, environmental objectives are central to transitioning the sector to be in line with global environmental goals, it may be useful to define separate policy objectives for CAP subsidies. This will reduce the ambivalent effects of agricultural subsidies, align subsidies with Article 2(1)(c) PA and ensure that public money is spent for public goods (Section 3.2.1). Furthermore, it seems useful to integrate income provision into the general social welfare networks of the Member States. However, amending the objectives of the CAP requires a unanimous decision by the European Council, which appears unlikely. This is where civil society might come into play to exercise pressure on policy makers to push subsidy removal and reform.

### 4.2. Data and Transparency

Reforming agricultural subsidies depends on reliable reporting, data and transparency. At the international level, the Paris Agreement incorporates a framework for transparency of support (Article 13(6)). However, this framework lacks robust accounting rules as to what qualifies as climate finance [168]. At the WTO, the Agreement on Agriculture also includes a notification requirement. Members of the WTO have to submit information about agricultural subsidies to the WTO. However, compliance is limited. Approximately one quarter of the notifications between 1995 and 2019 are outstanding [169]. To address this issue, the Members have submitted different proposals to the WTO [34,169]. It remains to be seen if these proposals will be adopted and improve the level of notifications.

At the EU level, data on agricultural State aid is frequently insufficient. For example, the ABER Regulation only applies to micro, small and medium-sized enterprises active in the agricultural sector (Article 1(1)(a)). These enterprises are defined by the number of employees, annual turnover and annual balance sheet. Identifying the number of agricultural holdings which qualify as micro, small and medium-sized enterprises is impossible. Neither the Commission nor eurostat have suitable data. Instead, eurostat uses the number of hectares per farm to measure the size of farms rather than their annual turnover or balance sheet [170]. In 2015, the Commission adopted a strategy for agricultural statistics for 2020 and beyond to improve data quality and coherence [171]. However, in a fitness check of the of the State aid modernisation, the Commission continues to point to flawed data and a lack of transparency [172]. Better data and transparency are also required for CAP subsidies as there are gaps in the reporting of Member States and harmonised data at the EU level is missing, which makes assessing CAP subsidies difficult [129]. Data quality and transparency are an ongoing challenge which require more effective instruments.

### 4.3. Subsidies and Other Policy Instruments

The goals of the Paris Agreement and the CBD require net zero emissions in a very short time, stopping biodiversity loss and restoring biodiversity. Subsidies are currently pervasive in agriculture. In previous publications, we proposed two emissions trading schemes with caps aligned to the goal of the Paris Agreement [12,13,173]. To limit the use of fossil fuels, we propose reforming the current EU emissions trading scheme into an emissions trading scheme with a more ambitious cap in line with Article 2(1) PA, the inclusion of fossil fuel usage in every sector, and the removal of certificates from earlier trading periods. A second emissions trading scheme for livestock products will not only reduce emissions but contribute to the goal of the CBD by reducing livestock numbers. These instruments have to be combined with other willing states on a global scale ('climate club'), and border adjustments targeting states without similar policy approaches. Additional command-and-control measures such as a livestock-to-land ratio are required to address, for example, nutrient hotspots and the contamination of mineral fertilisers with heavy metals, measures which also benefit biodiversity including soil biodiversity [12,13] (see [173] for more detail). These policy instruments effectively target the drivers of climate change and biodiversity loss. In addressing easily measurable governance units on a broad

geographical scale, they would avoid typical governance problems such as shifting effects, rebound effects, or enforcement problems. Furthermore, they take various motivational factors into account such as self-interest, conceptions of normality, path dependencies, and problems of collective goods. Moreover, these instruments are freedom-friendly and address technological and behavioural change at relatively low economic costs, which is the best-known argument for emissions trading schemes.

In contrast, subsidies are unable to cut, for example, fossil fuels to zero in a timely manner because subsidies do not effectively address the drivers of unsustainability. Furthermore, adopting subsidies to achieve the same emissions reduction would be more expensive compared with other market-based instruments such as cap-and-trade schemes because they are less market-oriented and, therefore, less cost-efficient [42,174,175]. Moreover, caps and levies have a broader scope than subsidies since they usually address, for example, the acquisition and efficient use of products. Social distribution issues arise with caps and levies as much as with subsidies (Section 3.1.2).

Subsidies, however, remain a useful complementary instrument to remunerate the provision of certain public goods as long as they are constructed in a way that they do not, or to the smallest extent possible, suffer from typical governance problems. For example, the proposed emissions trading scheme for livestock products reduces livestock numbers. Therefore, a broad provision of subsidies for the extensification of livestock production does not seem useful. Instead, biodiversity restoration requires subsidies as it is not covered by the proposed instruments, although reducing pressure on land by the proposed instruments could go far. In doing so, subsidies would be allocated to the provision of public goods (Section 3.1.2). Under these circumstances, it is conceivable to provide targeted subsidies for livestock to encourage farmers to keep animals where it is not economically attractive but enhances biodiversity. In restoring biodiversity, these targeted subsidies would frequently also be beneficial for combating climate change. Besides, given biodiversity restoration is highly effective when adopted at the landscape level, collaborative subsidies appear promising [128,139,176,177], which also reduces shifting effects. Multiple studies further highlight the importance of adapting subsidies to local conditions to achieve high environmental outcomes (Section 3.1.3). All of these subsidies have to be highly targeted with regard to global environmental goals and require robust minimum standards to ensure they are not watered down [106,178]. Still, biodiversity impacts might take a long time to realise (Section 3.1.3). As small-scale instruments, these subsidies will continue to suffer from enforcement problems and monitoring issues, although they might be smaller than under the current CAP as overall fewer subsidies will be provided. These challenges again emphasize that subsidies are not effective in addressing the drivers of global environmental challenges and have to be downscaled substantially.

Developing, implementing and monitoring these targeted subsidies requires expert knowledge [179], bureaucratic capacities, and knowledge transfer to farmers where needed [130,131,180]. Bringing the CAP in line with global environmental goals would free up substantial resources in public administration (e.g., by reducing monitoring of income support subsidies) and make funds available through the elimination of income support. These resources could be used to develop and implement targeted subsidies by, for example, extending agri-environment-climate commitments of the second pillar of the CAP. Moreover, making agri-environment-climate commitments more effective and developing new subsidies require research. This includes, for example, research on results-based subsidies to, for example, identify further suitable and practical proxies to monitor results-based payments as a promising alternative to dominant action-based subsidies of the CAP. Besides, further developing alternative fertilisers such as sewage sludge incinerator ash demands studies to assess their fertilising potential, for example [181]. In spite of this, CAP spending into research and innovation accounted for just 0.4% of CAP expenditure in 2020 [6]. To be able to combat climate change and biodiversity loss, more subsidies into R&D are urgently needed.

Subsidies have to be in line with WTO rules. Where subsidies provide farmers more than additional costs and income forgone, such as results-based subsidies, they qualify as Amber Box subsidies. Above a certain threshold, these subsidies have to be reduced. However, the Amber Box subsidies of the EU are far below this threshold [182]. Therefore, WTO rules do not generally hinder the adoption of subsidies which provide farmers with more than additional costs and income forgone [41].

Removing and reforming agricultural subsidies to be in line with global environmental goals offers various additional advantages. Firstly, subsidies are expensive. Reforming subsidies and removing environmentally harmful subsidies makes public funds available which can be used to effectively combat climate change and biodiversity loss. Secondly, reforming agricultural subsidies holds potential to reduce income inequality, although the impacts may not be clear cut [183–186]. However, most of all, removing environmentally harmful subsidies and implementing targeted subsidies limits pressure on climate and biodiversity and instead contributes to global environmental goals.

## 5. Conclusions

This article aimed to assess the extent to which (agricultural) subsidies can play a role in the transition to sustainability. Nutrient (phosphorus) management served as an example. Subsidies in agriculture are currently pervasive and they are only one puzzle piece in transitioning the agricultural sector to be in line with global environmental goals. The governance analysis highlights that—compared with other policy instruments—subsidies are less effective at achieving these goals. Therefore, agricultural subsidies should be implemented only as complementary instrument to provide public goods and R&D. Likewise, environmental agreements require removing environmentally harmful subsidies and implementing subsidies to achieve global environmental goals. All of this implies that agricultural subsidies have to be drastically downscaled. Besides, environmental objectives need to be streamlined through EU law to ensure agricultural subsidies are in line with global environmental goals and to eliminate less visible subsidies. The core ideas of (EU) agricultural policy need to be reformed and data and transparency enhanced.

**Author Contributions:** K.H.: Conceptualisation, Methodology, Formal Analysis, Investigation, Writing—Original Draft, Writing—Review and Editing; F.E.: Methodology, Writing—Original Draft, Supervision; L.S.: Formal Analysis, Writing—Original Draft; P.R.: Formal Analysis. All authors have read and agreed to the published version of the manuscript.

**Funding:** This research was funded by the Leibniz Association within the scope of the Leibniz ScienceCampus Phosphorus Research Rostock and by the German Federal Ministry of Education and Research (BMBF) within the BonaRes project InnoSoilPhos (No. 031B1061A).

**Institutional Review Board Statement:** Not applicable.

**Informed Consent Statement:** Not applicable.

**Data Availability Statement:** Not applicable.

**Acknowledgments:** Gábor Padisák from DG Competition for helpful comments on agricultural statistics of the EU, Theresa Rath (Research Unit Sustainability and Climate Policy) for feedback on the legal framework of subsidies. Beatrice Garske (Research Unit Sustainability and Climate Policy) for proofreading and feedback.

**Conflicts of Interest:** The authors declare no conflict of interest.

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
