# Peer review of "Potentials and Limitations of Subsidies in Sustainability Governance: The Example of Agriculture"

_sustainability, doi:10.3390/su142315859_

Round 1
Reviewer 1 Report
The paper addresses an important and timely issue. It is also well covered.
It is confusing to read how the cited works are used, for instance some are straightforward [27-29], some are extra words (article, or para)?
Why are so many brackets in the paper? Are the texts (e.g.) in the those brackets important, for instance (agricultural) is used many times throughout the manuscript.
Line 121 [13, 18] and line 132 [e.g.23, 24]: How are the references used to support the statement?
Line 141: 'theoretical and methodical': what do the authors mean, theoretical or methodical?
Reviewer 2 Report
The paper is potentially interesting, but it needs substantial revision before publication. The motivation of the paper should be improved, it is not clear what is the contribution to the existing literature. I do not really understand what a "broader perspective" (p. 2. line 54) means. Honestly, I do not see the story of what you like to sell to your potential reader. I do not see the value added to the qualitative governance analysis to any impact evaluation methodology. There is a vast literature on the effectiveness on impacts of the Common Agricultural Policy, including various environmental schemes, however, the paper fully ignores the knowledge of this literature, especially in the context of research question 4.
The presentation of the paper is extremely boring, it was very difficult to maintain my interest to reach the end of the paper. I do not think that I am alone in this respect. After reading the paper I asked myself what can I learn from this paper, but I had not a clear answer. The paper includes an extremely lot of information without a readable and understandable way. For example on page 8, in the last paragraph you list the lot percentage number (5), I guess nobody will remember the first percentage when one reaches the end of the paragraph.
The conclusions simply reinforce what we know from agricultural economics literature for a long time. These criticisms are well known at least 30 years.
Reviewer 3 Report
This article is very well written and tackles a hotly debated issue, the link between agricultural subsidies and environmental impacts.
My main comment is that some statements should be more nuanced, and limits to the analysis should be underlined with more discussion in section 4:
- The article focuses on nutrient management. In the conclusion, a short discussion should be provided on how the analysis would differ when the case of pesticides is taken. For example, the article often claims that agriculture can both harm and benefit the environment, and provides an analysis in this sense. But do pesticides benefit the environment?
- The analysis is based on the hypothesis that the impact of subsidies is immediate and that environmental changes are immediate. However, environmental changes take time, as well as the impact of subsidies on such changes. It should be discussed in section 4.
- The article is based on the hypothesis that agricultural subsidies have only negative impact, and income subsidies in particular should be fully removed, on the ground that ‘earning an adequate income is not a public good’. This should be less strong and better discussed and nuanced. Decent income for farmers means that farms are maintained, and this maintenance of farms is the public good. Studies show that society wants farms and agriculture, even if farms do not produce agricultural production. Citizens are even willing to pay for maintaining farm holdings. The reasons are aesthetics (landscape), cultural (maintaining traditions), employment…, that is to say social sustainability. And income support is a means to achieve this social sustainability. This should be very well balanced with the objective of environmental sustainability advocated in the article, and the message of the article should be toned down in this respect.
Minor comments:
- Lines 44 ‘these subsidies fail to ensure food security, farm livelihoods and sustainable practices’: this statement is too strong, it needs to be rephrased (e.g. ‘In general’ or ‘On average’).
- Lines 146-147 ‘The second pillar focusses on rural development primarily through e.g., voluntary multi-annual programs’: give precisions and examples on these programmes.
- It may be valuable to integrate the following article in the discussion:
Karmen Erjavec, Emil Erjavec. 2015. ‘Greening the CAP’ – Just a fashionable justification? A discourse analysis of the 2014–2020 CAP reform documents. Food Policy, Volume 51, Pages 53-62, https://doi.org/10.1016/j.foodpol.2014.12.006
- Lines 199-200 ‘Likewise, the EU provides subsidies to farmers to reduce nutrient inputs into the environment’: give precisions and examples on these subsidies.
- On Figure 2 grey boxes are too dark: when printed on paper, the text inside these boxes is not readable.
- Lines 315-316: explain what a ‘rebound effect’ is.
- Line 382: explain what ‘undercomplexity’ is.
- Lines 411-412: explain what ‘couples and decoupled subsidies’ are.
-
Round 2
Reviewer 1 Report
The authors have responded the reviewers comments well.
I would suggest the manuscript to be proofread by a professional editor.
Reviewer 2 Report
My comments were partly addressed. Other parts were ignored. I am still not convinced that the paper is publishable.
Reviewer 3 Report
There are still remaining issues that I highlighted in the previous review round and that have not been addressed sufficiently.
(1) Lines 44-45: My comment was that the statement ‘these subsidies fail to ensure food security, farm livelihoods and sustainable practices’ is too strong, and needs to be rephrased (e.g. ‘In general’ or ‘On average’).
The authors added ‘In general’ to the sentence that follows (‘In general, agricultural subsidies affect environmental outcomes by changing how much is produced, which products are produced, where these products are produced, and how they are produced’) but that was not the sentence that I pointed.
(2) The authors indicate that the case of pesticides is a case in itself and did not include any sentence related to this. I agree with the authors that this is an article in itself, however, the authors should add a sentence (e.g. in conclusion) recalling that the article focuses on nutrient management, and that the conclusions might be different for other cases e.g. pesticides and that this is an avenue for future research.
(3) Regarding my comment on the sentence ‘Earning an adequate income is not a public good [122]’, I understand the point of view of the authors but still they should mention the other point of view, and also note that the literature quoted there ([122]) is not an academic publication and relates only to Germany. It could be a simple mention e.g. ‘Although there may be public benefits in the maintenance of farms, earning an adequate income is not a public good [122]’.
